# Cost-Effectiveness of an Organized Lung Cancer Screening Program for Asbestos-Exposed Subjects

**DOI:** 10.3390/cancers14174089

**Published:** 2022-08-24

**Authors:** Sébastien Gendarme, Jean-Claude Pairon, Pascal Andujar, François Laurent, Patrick Brochard, Fleur Delva, Bénédicte Clin, Antoine Gislard, Christophe Paris, Isabelle Thaon, Helene Goussault, Florence Canoui-Poitrine, Christos Chouaïd

**Affiliations:** 1Pneumology Department, CHI Créteil, Inserm U955, UPEC, IMRB, F-94000 Creteil, France; 2Occupational Diseases Department, CHI Créteil, Inserm U955, UPEC, IMRB, F-94000 Creteil, France; 3Cardio-Thoracic Research Center of Bordeaux, INSERM U1045, University of Bordeaux, F-33000 Bordeaux, France; 4Occupational Diseases Department, CHU Bordeaux, F-33404 Bordeaux, France; 5Bordeaux Population Health Research Center, Inserm UMR1219-EPICENE, University of Bordeaux, F-33404 Bordeaux, France; 6Occupational Diseases Department, CHU Caen, F-14033 Caen, France; 7Occupational Diseases Department, CHU Rouen, F-76031 Rouen, France; 8Occupational Diseases Department, CHU Rennes, F-35033 Rennes, France; 9Occupational Diseases Department, CHRU Nancy, University of Lorraine, F-54000 Nancy, France; 10Public Health Department & Clinical Research Unit (URC Mondor), Henri-Mondor Hospital, Assistance-Publique Hôpitaux de Paris (AP-HP), F-94010 Creteil, France

**Keywords:** lung neoplasms, screening, occupational diseases, asbestos, cost-effectiveness

## Abstract

**Simple Summary:**

Lung cancer screening experiments in smokers are underway in Europe, and data in populations with other risk factors for lung cancer, such as asbestos exposure, are expected. Our original article yielded a cost-effectiveness analysis of a lung cancer screening program in a population exposed to asbestos, based on the data from National Lung Cancer Screening trial and a French asbestos-exposed cohort (ARDCO cohort). Individual data from 14,218 subjects in the ARDCO cohort, followed for 20 years (2002–2022), have allowed several screening models to be established according to exposure level, smoking status and presence of radiological signs of asbestos exposure. For the whole cohort, an annual screening programme is not cost-effective, while screening every 2 years for smokers with high asbestos-exposure and subjects with asbestosis is cost-effective. This analysis has never been reported in the literature and could help in the establishment of inclusion criteria for future experiments in this population.

**Abstract:**

**Background**: The National Lung Screening Trial (NLST) and NELSON study opened the debate on the relevance of lung cancer (LC) screening in subjects exposed to occupational respiratory carcinogens. This analysis reported the incremental cost-effectiveness ratios (ICER) of an organized LC screening program for an asbestos-exposed population. **Methods:** Using Markov modelization, individuals with asbestos exposure were either monitored without intervention or annual low-dose thoracic computed-tomography (LDTCT) scan LC screening. LC incidence came from a prospective observational cohort of subjects with occupational asbestos exposure. The intervention parameters were those of the NLST study. Utilities and LC-management costs came from published reports. A sensitivity analysis evaluated different screening strategies. **Results:** The respective quality-adjusted life year (QALY) gain, supplementary costs and ICER [95% confidence interval] were: 0.040 [0.010–0.065] QALY, 6900 [3700–11,800] € and 170,000 [75,000–645,000] €/QALY for all asbestos-exposed subjects; and 0.144 [0.071–0.216] QALY, 13,000 [5700–26,800] € and 90,000 [35,000–276,000] €/QALY for smokers with high exposure. When screening was based on biennial LDTCT scans, the ICER was 45,000 [95% CI: 15,000–116,000] €/QALY. **Conclusions:** Compared to the usual ICER thresholds, biennial LDTCT scan LC screening for smokers with high occupational exposure to asbestos is acceptable and preferable to annual scans.

## 1. Introduction

In 2018, lung cancer (LC) was the most common newly diagnosed malignancy, with 1.8 million new cases worldwide [1]. It was also the leading cause of cancer-attributed mortality, responsible for approximately 20% of cancer deaths [2]. In addition to smoking, occupational exposure to respiratory carcinogens constitutes another major risk factor for LC. The International Agency for Research on Cancer has classified several agents and exposure contexts as definitely carcinogenic [3]. Among occupational exposures, asbestos is the main risk factor for LC. The fraction of LCs attributable to asbestos is estimated to be between 5% and 20% [4,5]. In France, post-professional monitoring with thoracic computed-tomography (TCT) scans every 5 to 10 years, depending on the level of exposure, is recommended for asbestos-exposed subjects [6,7]. However, those recommendations also specify that the benefit of TCT scan screening for malignant pathologies has not yet been demonstrated.

Regardless of the risk factor and despite the therapeutic advances of the last few years, LC prognosis is poor, notably because of its often late diagnosis. Screening by low-dose TCT (LDTCT) can shift LC detection to an earlier stage and decrease LC mortality in high-risk individuals [8,9]. The proportion of lung cancer detected at a localized stage turns out to be a key parameter to evaluate the benefits of LC screening programs [10], as it is clearly associated with the reduction of LC mortality [11,12,13]. That benefit is generally attributed to surgical treatment resulting in curative resection of the tumor. To date, the relevance of a screening program is based on the results of several randomized clinical trials in smokers or former smokers [8,9]. The American National Lung Screening Trial (NLST) [8] included more than 53,000 smokers and former smokers (defined as having stopped for at least 15 years) with >30 pack-years, 55–74 years old and compared the efficacy of 3 scans at 1-year intervals and additional 2-year follow-up with chest X-ray. However, that study did not take into consideration occupational exposure. Subjects in the LDTCT scan arm had, respectively, 20% and 6.7% fewer LCs and lower overall mortality. The Dutch–Belgian LC screening (NELSON) study, with organized screening of populations with the same LC risk factors, i.e., age and smoking, also showed 24% less LC mortality, without any overall mortality difference, but samples sufficiently large to show an effect on the latter was not planned [9].

Those observations opened the debate on the opportunity offered by an LC screening program for persons at risk of LC linked to other risk factors, e.g., smoking and occupational exposure to lung carcinogens, especially asbestos [14].

A systematic meta-analysis reviewing all cohort studies involving TCT scan screening of former asbestos-exposed workers showed that such LC detection rates among exposed workers were at least equal to that for heavy smokers [15]. TCT scan screening of asbestos-exposed workers apparently accurately detected asymptomatic LCs and identified a percentage of stage I cancers similar to that for smokers. The authors concluded that TCT scan screening of asbestos-exposed workers could contribute to lower mortality, similar to that observed for heavy smokers, and, therefore, should not be ignored, particularly for subjects co-exposed to tobacco. Another meta-analysis showed that asbestos-exposed persons had an LC detection rate [95% confidence interval (CI)] of 0.94% [95% CI: 0.47–1.53] for smokers and 0.11% [0.00–0.43] for never-smokers [16]. Indeed, the LC risk varied according to the level and duration of asbestos exposure, the quantity of tobacco smoked and other factors linked to exposure, including the presence of pleural plaques [17] or asbestosis [18]. Taking these different factors into account needs to be evaluated to better target the population for which LC screening would be the most relevant, particularly in medical–economic terms.

This analysis was undertaken to modelize the cost–efficacy relationship for organized LC screening for subjects with occupational asbestos exposure. 

## 2. Materials and Methods

The analysis relied on Markov modelization to compare, for persons with occupational asbestos exposure, monitoring without intervention or LDTCT scan screening for LC. The non-intervention–strategy group’s clinical information and care use came from the prospective observational ARDCO cohort of subjects with occupational asbestos exposure [17,19,20]. 

As previously reported [21], that cohort was constituted of retired or unemployed workers with a history of occupational asbestos exposure in four French regions. They underwent a free medical check-up, which included TCT scan and pulmonary function tests [20]. Industrial hygienists used standardized questionnaires to evaluate asbestos exposure based on each subject’s complete work history, which enabled their classification as having low, intermediate or high exposure. The other characteristics collected were: age, sex and smoking status at cohort entry. The participants were followed for LC diagnosis and vital status until July 2019. Underlying and contributing causes of death, as stipulated on death certificates, were obtained. Respiratory-targeted care use was obtained from the French National Health Insurance (FNHI) data and self-reported questionnaires completed at regular intervals.

Markov modelization comprised five health states: “subject in good health”, “subject with localized LC”, “subject with disseminated LC”, “false-positive screening” and “died”, and seven transitions between states (Figure 1).

The main analysis examined the strategy based on the NLST intervention applied to the entire cohort of asbestos-exposed individuals, regardless of their level of smoking exposure. The efficacy results of that trial, adjusted for age, were applied to ARDCO cohort participants to determine the numbers and their stages of LCs that would have been diagnosed in this cohort if the NLST screening strategy had been applied. The model cycle was 1 year, with a temporal horizon of an entire lifetime. Efficacy is reported as an incremental of the number of years of life gained adjusted for quality of life, i.e., quality-adjusted life years (QALYs). The confidence intervals (CIs) were estimated with the Monte Carlo method, taking the minimum and maximum values from among the 10,000 independent draws from input parameter distributions. In each simulation, the entire ARDCO cohort is given a random cost for each procedure according to a gamma distribution, random utilities according to a beta distribution and random transition probabilities to localized or disseminated LC according to a normal distribution. The modelization relied on the following hypotheses: the NLST results obtained for a population of smokers and former smokers are transposable to a population having an occupations risk (15); the control arm of that trial corresponds to the non-intervention strategy currently proposed in France to monitor subjects with occupational exposure to respiratory carcinogens; the LDTCT scan performance is independent of time; and screening does not modify the life expectancy of subjects without an LC diagnosis. Conceptually, the net benefit of the intervention (LDCT scan screening) is determined in our modelization by four key parameters: earlier detection of localized cancers (parameter A), a decrease in the number of LC diagnosed at an advanced stage (parameter B) and a benefit in terms of life expectancy related to stage-specific mortality (parameter D and E). From the clinical side, several outcomes are expected in subjects in the “Localized LC” state. The gain in life expectancy is variable across subjects: either a return to a “healthy” state in case of curative treatment, a progression to diffuse cancer, or death. As such, data were not available, and in order to simplify the model, all three relevant transitions are considered simultaneously in one transition to the death state (parameter D) that reflects those different outcomes. The probabilities of death for subjects in good health according to age (parameter F) were derived from mortality data from the French National Institute of Demographic Studies (INED) [22]. The probabilities of transition to localized and disseminated LC (parameter A and B) were taken from the NLST study for the screening strategy [8] and from a French prospective registry for the usual care strategy [23]. The probabilities of death as a function of LC stage (parameter D and E) were taken from a French prospective registry study (KBP cohort) [23]. The KBP cohort included 7051 French patients with LC, accounting for about 20% of all cases of lung cancer in 2010, enabled comparison and extrapolation to the French population of lung cancer patients. Transition to death was then derived from background survival probability, to which an LC stage-specific hazard ratio for death (parametric distributions) was applied [24]. The probability of false-positive findings came from the NELSON study [9]. Preliminary data based on studies conducted in France indicated a false-positive rate similar to that of the NELSON study [25]. We evaluated the NLST false-positive rate in a sensitivity study. Utilities are taken from the literature [26], without disutility for subjects with “false-positive–screening findings” [27].

The cost analysis is limited to direct costs, from the FNHI perspective. For the non-intervention strategy, the costs of hospitalizations and respiratory-targeted out-of-hospital care were taken into consideration. For the screening strategy, the costs of subject selection (occupational medicine consultation), screening examinations (LDTCT scan and consultation with a pneumologist), examinations engendered by the false-positive–screening findings and organizing the screening were taken into account. The costs of managing LC in both arms were established based on the literature data, differentiating between localized (stage I and II) [28,29,30] and metastatic forms [31]. Costs are reported with a 3% discount rate [32].

The sensitivity analysis examined different assumptions according to structural, methodological and model parameters uncertainty [33]. The ICERs for the different subgroups defined by smoker status at entry into the program, intensity of asbestos exposure and the presence of pleural plaques or asbestosis were provided. Uncertainties in the model’s main transition parameters were analyzed, with the NLST study’s false-positive rate for parameter C, the NELSON study’s LC stage shift for parameter A and B, and the KBP 2010 study’s extreme values of death probabilities for parameter D and E as maximum constraints. That analysis also sought the impact at a temporal horizon of 10 years, discount rates of 2% and 6%, overdiagnosed rates of 3–23%, impact of the costs of subject-initiated care use and false-positive–screening engendered additional investigations and the quality-of-life decline resulting from unnecessary examinations. Finally, sensitivity analysis also addressed an intervention strategy based on a biennial LDTCT scan. The latter reproduced the strategy implemented in the Multicentric Italian Lung Detection (MILD) trial, applying a 2-year cycle in the model and taking into account the probability of being diagnosed with LC after negative screening (i.e., the cancer interval) [34,35]. The impact of the number of between-screen LC diagnoses is reported in the sensitivity analysis. All analyses were computed with Microsoft® Excel® 2016 MSO (Microsoft Corporation, Paris, France) software and R Studio (version 4.1.2, Boston, MA, USA) software for the graphic representation of the sensitivity analysis. 

## 3. Results

### 3.1. Population Characteristics and Parameters

The characteristics of ARDCO cohort and NLST participants are reported in Table 1. The ARDCO cohort enrolled a higher percentage of men (94.8% vs. 59%) and lower percentage <60 years old (23.4% vs. 42.8%) than the NLST study. The respective low, intermediate or high occupational asbestos-exposure rates of ARDCO cohort participants were 7.5%, 68.0% and 24.5%, whereas respiratory carcinogen exposure (notably asbestos) was not yielded in the NLST. The ARDCO cohort included 70% smokers or former smokers.

Based on a homogeneous distribution of the number of incident cases over time, LC incidence was 2.3 for 1000 person-years for the entire ARDCO cohort population and 7.1 for 1000 person-years for the subgroups of smokers with high asbestos exposure (Table 2). The incidence rate after implementation of the screening strategy was evaluated based on the relative risk of 1.13 [95% CI: 1.03–1.23], i.e., 13% additional cancers diagnosed. All the parameters included in the model are summarized in Table 3.

### 3.2. Main Analysis

The LC screening program’s annual incremental cost [95% CI] for all asbestos-exposed subjects >55 years old for a lifetime time horizon and with a discount rate of 3% was 6900 € [3700–11,800 €] per person, for a QALY gain [95% CI] of 0.040 [0.010–0.065] per person, i.e., an ICER [95% CI] of 170,000 [74,000–645,000] €/QALY (Table 4 and Scatter plot shown in Appendix A). For smokers with high asbestos exposure, the cost [95% CI] per person was 13,000 € [5652–26,800 €], with a QALY gain of 0.144 [0.071–0.216], i.e., an ICER of 90,000 [35,000–276,000] €/QALY (Table 5). When screening was based on LDTCT scans every 2 years, the ICERs were 64,000 [20,000–143,000] €/QALY and 45,000 [15,000–116,000] €/QALY, respectively, for the entire population and the subgroup of smokers with high exposure.

### 3.3. Sensitivity Analysis

Sensitivity analysis of that subgroup showed that the interval between LDTCT scans was the parameter that generated the most important ICER improvement (Figure 2). Application of the management algorithm used in the NELSON study for nodules diagnosed by LDTCT scan, notably based on their volumetric doubling time, by limiting the false-positive rate, was able to lower the ICER to 45,000 €/QALY, compared to the strategy implemented in the NLST (false-positive rate: 23.3%; ICER: 65,000 €/QALY). The main model’s transition parameters have a limited impact; when the rate of localized and disseminated LC found in the NELSON study is applied (51.2% and 48.8%, respectively), the ICER is 54,000 €/QALY. Finally, lowering the quality of life by a factor of 0.10 in false positives had little impact on the model’s cost/utility ratio, with an ICER of 48,000 €/QALY. For an acceptability threshold of 50,000 €/QALY, this strategy was cost-effective for 58.5% of the modelized simulations (Appendix A). For an acceptability threshold of 60,000 €/QALY, the eligible population could be extended to both smokers and never-smokers with high exposure. The ICERs of the different screening strategies according to the participants’ characteristics, i.e., age, asbestos-exposure level, smoking status and radiological anomalies linked to that exposure are reported in Table 5 and Appendix A.

## 4. Discussion

The medical–economic impact of LC screening programs for smokers in the general population, addressed in numerous studies as a function of the risks and healthcare systems, yielded ICERs ranging from 28,000 to 169,000 $/QALY in North America [36,37,38], 44,000 $/QALY in New Zealand [39], and from 35,674 to 69,099 €/QALY in Europe [40]. Data on persons exposed to asbestos are scarce [41].

Our modelization results showed that when only occupational exposure was considered, annual LC LDTCT scan screening was not cost-effective for most western countries, with an ICER of 170,000 €/QALY. In contrast, biennial LDTCT scan screening, when limited to the subgroup of smokers with high asbestos exposure, yielded an ICER of 45,000 €/QALY, which could be considered acceptable in most settings [42]. Depending on the willingness-to-pay of the different healthcare systems, LDTCT scan screening every 2 years could also be extended to include smokers, regardless of the level of exposure, and persons with asbestosis. Evaluation of the degree of occupational exposure is another important eligibility factor. Another analysis based on biennial LDTCT scans of subjects with occupational exposure [41] found an ICER that ranged from 25,400 to 41,400 $/QALY. According to that analysis, the optimal population would be smokers with >15 pack-years and an occupational exposure-linked relative risk >2 [20].

Based on our modelization, an organized biennial LDTCT scan screening for all individuals exposed to asbestos, without taking smoking exposure into account, does not seem to be cost-effective. That finding is in agreement with the results of other studies [16,43]. To target a population at greater risk, Markovitz et al. [44] recommended LC screening for subjects exposed to asbestos for at least 5 years, and smoking with >10 pack-years with no limitation to the length of time as a former smoker, asbestosis, family history of LC, chronic pulmonary disease, prior cancer or an occupational co-exposure. Welsh et al. [45] proposed LC screening for smokers exposed to asbestos, regardless of the number of pack-years of tobacco use.

One of the strengths of our modelization for the non-intervention strategy is the availability of prospective data from a cohort comprising >14,000 participants with occupational asbestos exposure that enabled estimation of LC incidence based on individual data, and the analysis of subgroups according to the level of occupational exposure, smoking status and presence of asbestos-linked imaging anomalies. The modelization also took into account asbestos-exposed subject-initiated use of respiratory-targeted care.

One of the study limitations is the comparability of the populations, especially the differences in smoking status, occupational exposure and sex between smokers in the general population of the NLST from which the efficacy criteria were taken and the asbestos-exposed ARDCO cohort participants. Nonetheless, the LC incidence in the NLST control arm is close to that of ARDCO cohort smokers (5.72 vs. 6.04 for 1000 person-years, respectively) [8]. Another limitation is that the cost analysis was restricted to direct costs, not taking into account, most particularly, the indirect costs linked to overdiagnosis, potential radiation-induced cancers and false-positive morbidities. Regarding the model design, it would be suitable to make a distinction between several clinical outcomes for subjects with localized LC, which would provide a more accurate model reflecting the variability of outcomes across subjects. For the sake of simplicity and due to lack of data, we assumed that all patients in the localized LC state will result in the same probability to death. Lastly, the lack of French data on health utility for subjects with LC should be mentioned [26].

## 5. Conclusions

In light of the usual ICER thresholds applied, biennial LC LDTCT scan screening for smokers with high occupational asbestos exposure is acceptable and preferable to annual LDTCT scan screening. The results of an ongoing prospective clinical trial [46] are expected to validate the modelization results reported herein.

## Figures and Tables

**Figure 1 cancers-14-04089-f001:**
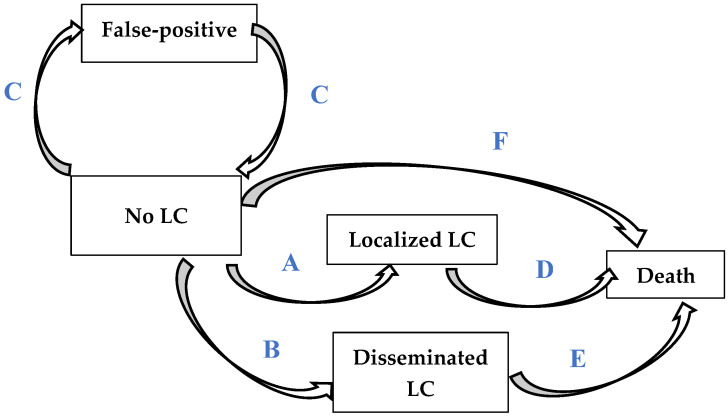
Structure of the Markov model (cycle = 1 year). A: Probability of localized (stage I and II) lung cancers (LCs) with (8) or without screening (23). B: Probability of disseminated (stage III and IV) LCs with (8) or without screening (23). C: Rate of false-positive computed-tomography scan findings (9). D: Probability of death attributable to localized LCs (23). E: Probability of death attributable to disseminated LCs (23). F: Probability of death without LC (22).

**Figure 2 cancers-14-04089-f002:**
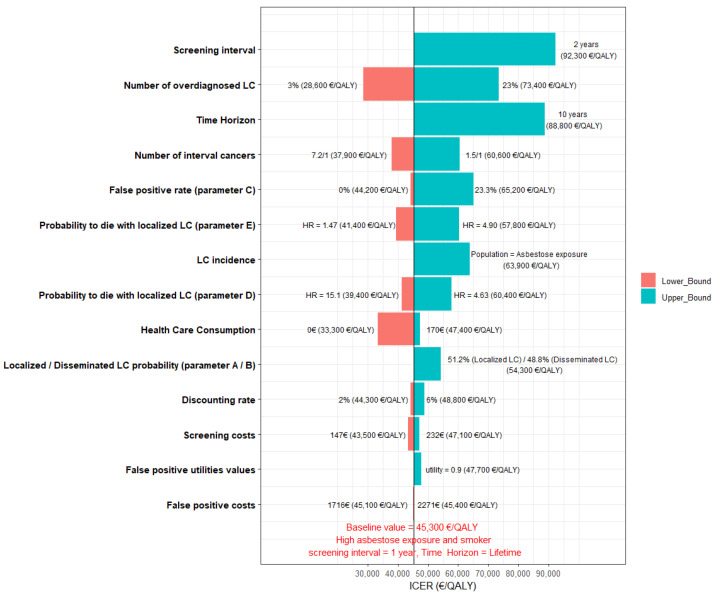
Tornado diagram of the sensitivity analysis representing the variability of ICER according to each parameter uncertainty, with baseline value corresponding to the strategy “scan every 2 years from 55 years of age for smokers with high exposure to asbestos, lifetime horizon”. LC, lung cancer; ICER, incremental cost-effectiveness ratio; QALYs, quality-adjusted life years.

**Table 1 cancers-14-04089-t001:** Sociodemographic and Clinical Characteristics of Subjects Included in the ARDCO Cohort and the NLST [8].

Characteristic	ARDCO	NLST	
	Cohort	CT Group	Controls
Number of subjects	14,218	26,722	26,732
Male sex, *n* (%)	13,481 (94.8%)	15,770 (59.0%)	15,762 (59.0%)
Age at inclusion, years			
<60	3332 (23.4%)	11,442 (42.8%)	11,424 (42.7%)
≥60 and <75	10,490 (73.8%)	15,279 (57.2%)	15,305 (57.3%)
≥75	396 (2.8%)	1 (<0.1%)	3 (<0.1%)
Smoker status at inclusion ^1^			
Never-smoker	2943 (20.7%)	0 (0%)	0 (0%)
Ex-smoker	6042 (42.5%)	13,860 (51.9%)	13,832 (51.7%)
Smoker	835 (5.9%)	12,862 (48.1%)	12,900 (48.3%)
Asbestos exposure			
Low	1070 (7.5%)	NA	NA
Intermediate	9660 (67.9%)	NA	NA
High	3488 (24.5%)	NA	NA

Abbreviation: NA, not applicable because asbestos exposure was not taken into account. ^1^ missing data for the ARDCO cohort: *n* = 4398 subjects.

**Table 2 cancers-14-04089-t002:** Lung Cancer Incidence According to Age at Inclusion in the Screening Program, Smoker Status, Asbestos Exposure and Benign Scan-Detected Asbestos-Linked Anomaly Status (ARDCO Cohort).

Parameter	Lung Cancer Incidence (Per 1000 Person-Years)			
	All Ages	<60 Years	60–75 Years	>75 Years
Total population	2.30‰	2.61‰	2.21‰	2.12‰
Smokers	6.04‰	6.41‰	5.57‰	NA
Smokers & former smokers	2.78‰	3.22‰	2.30‰	1.85‰
Asbestos exposure				
High	2.90‰	3.03‰	2.83‰	2.64‰
Intermediate	2.20‰	2.49‰	2.15‰	1.91‰
High and smoker	7.07‰	6.36‰	7.82‰	NA
Pleural plaques	2.31‰	2.42‰	2.25‰	2.80‰
Subjects with asbestosis	5.00‰	NA	6.06‰	NA

Abbreviation: NA, non-applicable.

**Table 3 cancers-14-04089-t003:** Main Analysis Model Parameters.

Parameter	Model Values			Probability	Ref
	Determinist	Low: Parametric	High: Parametric	Distribution	
Probability of transition between health states without screening					
Localized LC (A ^1^)	SIR ^2^ × 0.181	SIR ^2^ × 0.171	SIR ^2^ × 0.191	Normal	[23]
Disseminated LC (B ^1^)	SIR ^2^ × 0.819	SIR ^2^ × 0.809	SIR ^2^ × 0.829	Normal	[23]
Probability of transition between health states with screening					
Localized LC (A^s^) ^3^	SIR ^2^ × 0.702	SIR ^2^ × 0.694	SIR ^2^ × 0.710	Normal	[8]
Disseminated LC (B^s^) ^3^	SIR ^2^ × 0.298	SIR ^2^ × 0.290	SIR ^2^ × 0.306	Normal	[8]
HR overdiagnosis	1.13	–	–	–	[8]
Probability of false-positives (C ^1^)	1.2%	–	–	–	[9]
Model adaptation for a 2-year interval between scans					
LDTDT-detected LCs/cancer interval	2.8/13 ^4^	–	–	–	[9]
LCs detected every 2 vs. 1 year	1.5/1 ^5^	–	–	–	[35]
Probability of transition between health states (2 strategies)					
Death attributed to localized LCs (D ^1^)	HR: 2.68	–	–	–	[23]
Death attributed to disseminated LCs (E ^1^)	HR: 8.38	–	–	–	[23]
Non-LC death (F ^1^)	INED 2019 death table	–	–	–	[22], Appendix A
Costs					
Without screening ^6^	26 €	26 €	73 €	Gamma	Appendix A
With screening ^7^	189 €	147 €	232 €	Gamma	Appendix A
Localized LC					
- Surgical	13,390 €	6337 €	20,443 €	Gamma	[29]
- Post-surgical (/2 years)	19,057 €	16,770 €	21,429 €		[30]
Disseminated LC (/2 years)	33,132 €	29,357 € ^8^	34,305 € ^9^	Gamma	[31]
False-positives	2110 €	1716 €	2271 €	Gamma	Appendix A
Utility					
Localized LC	0.825	0.793	0.857	Beta	[26]
Disseminated LC	0.573	0.506	0.640	Beta	[26]
False-positives	1.000	0.970	1.000	Beta	[27]

Abbreviations: LC, lung cancer; SIR, standardized incidence ratio; HR, hazard ratio; INED, Institut National d’Etudes Démographiques (French National Institut for Demographic Studies). ^1^ Each capital letter A–F corresponds to the transition between health states indicated in Figure 1. ^2^ SIR corresponds to LC incidence, standardized for age, smoker status, asbestos exposure, pleural plaques and asbestosis (cf. Table 2). ^3^ An exposant s is added when the probability corresponds to the screening strategy. ^4^ For a 2-year between-scan interval, the NELSON study found 7.69 cancers detected for 1000 scans and 2.76 cancers per interval for 1000 scans for a ratio de 7.69/2.76 ≈ 2.8/1. ^5^ In the MILD study, LC incidences were 620 for 100,000 person-years in the annual screening arm and 457 for 100,000 person-years for the biennial scan arm, for a ratio of 1/0.75; therefore, to obtain this result, 1.5 times more LCs detected with biennial screening. ^6^ Non-intervention subject-initiated care use in the ARDCO cohort was estimated using two methods: data extracted from the codes for homogeneous patient groups (low value) for private hospitals and FNHI codes for respiratory-targeted interventions, or used responses to a questionnaire sent to ARDCO cohort subjects (high value). ^7^ Screening costs correspond to the expenditures engendered by: subject selection (occupational disease consultation), the low-dose thoracic computed-tomography scan, pneumology consultation and organizational costs/person. ^8^ Costs of disseminated cancers in elderly subjects according to McGuire et al. [31]. ^9^ Costs of disseminated cancers in young subjects according to McGuire et al. [31].

**Table 4 cancers-14-04089-t004:** Clinical outcomes, health system costs, QALYs gained and ICER for a screening intervention every year or every 2 years, starting at the age of 55, for the 14,218 subjects of ARDCO cohort and for an hypothetical population of 14,218 smokers with high asbestos exposure (time horizon: lifetime, discounted at 3% per annum, uncertainty intervals in parentheses estimated with the minimum and maximum values of the 10,000 Monte Carlo simulations).

Outputs	ARDCO Cohort Screening Every 1 y	ARDCO Cohort Screening Every 2 y	Smokers with High Asbestos Exposure Screening Every 1 y	Smokers with High Asbestos Exposure Screening Every 2 y
Number of subjects in the simulation	N = 14,218	N = 14,218	N = 14,218	N = 14,218
Localized LC: Usual care	169 (150–187)	169 (149–187)	513 (462–567)	513 (462–566)
Localized LC: Intervention scenario	740 (725–756)	641 (627–654)	2198 (2144–2248)	1931 (1897–1970)
Disseminated LC: Usual care	761 (743–777)	761 (745–777)	2304 (2256–2352)	2304 (2245–2358)
Disseminated LC: Intervention scenario	316 (300–331)	289 (274–302)	926 (874–969)	867 (824–912)
Total Number of False Positive results	4993 (4988–4998)	2551 (2550–2554)	4560 (4545–4574)	2357 (2351–2363)
Per capita				
Total Costs: Usual care (€)	5493 € (5136–7403)	5493 € (5121–7256)	15,264 € (12,931–19,454)	15,264 € (12,835–19,770)
Total Costs: Intervention scenario (€)	12,408 € (9049–18,811)	9653 € (6705–14,390)	28,310 € (19,518–45,422)	24,330 € (16,629–39,851)
Total QALYs: Usual care	17.6911 (17.6758–17.7066)	17.6911 (17.6757–17.7055)	17.1491 (17.1001–17.1951)	17.1491 (17.1063–17.1921)
Total QALYs: Intervention scenario	17.7314 (17.699–17.759)	17.7560 (17.7293–17.7798)	17.2931 (17.2064–17.3733)	17.3491 (17.2715–17.4256)
Total incremental cost (€)	6915 € (3671–11,774)	4161 € (1346–7542)	13,046 € (5652–26,757)	9066 € (2697–20,488)
QALYs gained	0.0403 (0.0094–0.0652)	0.0650 (0.0399–0.0879)	0.1440 (0.0717–0.2155)	0.2000 (0.1288–0.2601)
ICER (€/QALY)	171,575 €/QALY (74,669–644,761)	64,023 €/QALY (20,460–143,220)	90,624 €/QALY (35,405–276,018)	45,331 €/QALY (14,992–115,809)

Abbreviations: y, year; N, number; LC, lung cancer; €, euros; QALY, Quality-Adjusted Life Years.

**Table 5 cancers-14-04089-t005:** Incremental Cost Effectiveness Ratios (ICERs) for the Different Screening Strategies According to Population Characteristics (lifetime horizon and discount rate of 3% per annum).

Characteristic	Incremental Cost Effectiveness Ratio (€/QALY)					
	Annual Scan			Biennial Scan		
Age at screening start	50 years	55 years	60 years	50 years	55 years	60 years
Asbestos exposure						
Any	170,485	171,575	187,957	66,386	64,023	69,005
High	152,324	146,952	155,982	61,387	58,743	60,170
Intermediate	173,469	174,300	193,499	67,196	65,241	70,090
Any and smoker	117,769	114,854	117,955	52,179	49,195	51,099
High and smoker	103,039	90,624	90,809	47,661	45,331	41,597
Pleural plaques	167,606	157,823	157,215	65,916	60,790	61,333
Asbestosis	112,202	99,531	101,620	50,067	48,011	44,366

## Data Availability

The data that support the findings of this study are available on request from the corresponding author.

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
