# Peer review of "Cost-Effectiveness of an Organized Lung Cancer Screening Program for Asbestos-Exposed Subjects"

_cancers, 2022, doi:10.3390/cancers14174089_

Round 1

Reviewer 1 Report

This is an interesting cost-effectiveness study on a tumor that is rare but socially very relevant in specific geographycal areas where environmental or occupational exposure happened. Introduction well explain the reaserch question. Methods seems appropiate and well explained. Results are well reported and represented. Discussion is well structured and balanced. Conclusion correctly refers to mor reliable results that will come from an ongoing prospective trial

Author Response

This is an interesting cost-effectiveness study on a tumor that is rare but socially very relevant in specific geographycal areas where environmental or occupational exposure happened. Introduction well explain the reaserch question. Methods seems appropiate and well explained. Results are well reported and represented. Discussion is well structured and balanced. Conclusion correctly refers to more reliable results that will come from an ongoing prospective trial.

Thanks for this positive feedback

Reviewer 2 Report

PRELIMINARY COMMENT

In Table 3, the death rate resulting from localised LC expressed as a hazard ratio, denoted as C, is 8.38 (presumably in comparison with healthy subjects), while the death rate resulting from disseminated LC expressed as a hazard ratio, denoted as D, is 2.68 (presumably in comparison with healthy subjects). It is paradoxical that the death risk is higher for a localized disease than for a disseminated disease.

The present review is therefore based on the assumption that a trivial error has occurred and that the true values for this HR are 8.38 for disseminated LC and 2.68 for localised disease.

GENERAL COMMENTS

First of all, there are a couple of general, conceptual points that need to be clarified before examining the specific points raised by this study.

A successful screening of lung cancer (CR) implies that more diagnoses are made at an earlier stage. If there is no screening  at all, no increase in early diagnosis will occur. If the screening is made once yearly, more early diagnoses will be made. If the screening is made every two years, fewer early diagnoses will be made than in the case of yearly screening.

Each early diagnosis determines a benefit particularly when the early diagnosis  permits a curative resection of the tumour. The benefit is less relevant when early diagnosis only prolongs survival in the absence of a cure, because it is well known that, especially in the case of lung cancer, this prolongation can be limited to a few months and a fatal outcome is not avoided. By the way, the terms “early diagnosis” and “localized lung cancer” can be seen, more or less, as synonyms.

Hence, the crucial point with this analysis lies on the clinical side. What outcomes are expected after an early diagnosis (or localised disease)? Some patients (Case 1) will die all the same; others (Case 2) will experience some survival prolongation; others (Case 3) will benefit from a curative resection so that their residual life expectancy will equal that of a healthy population.  How were the simulated patients of the Markov model distributed across Cases 1, 2, and 3?

Likewise: What outcomes are expected from a disseminated disease? Likely, this distribution across Cases 1, 2, and 3 is considerably different.

If one examines this issue from a clinical side, its complexity seems to be overwhelming because, to my knowledge, no data (arranged exactly in this way) are available from the literature, and so another Markov study could be needed to provide an appropriate answer to these questions.

Nevertheless, the authors have provided an answer to this question by presenting the parameters denoted as A, B, C, and D (see Table 3).

In fact, in the simulations generated by the Markov model, there are four critical parameters:

  1. “A” (see Table 3): rate of Localised LC occurrence
  2. “B” (see Table 3): rate of disseminated LC occurrence
  3. “C” (see Table 3): death rate resulting from localised LC (Note: is should be noted that, in Table 3, this parameter [value= 8.38] is expressed as a hazard ratio, presumably in comparison with healthy subjects);
  4. “D” (see Table 3): death rate resulting from disseminated LC ((Note: is should be noted that, in Table 3, this parameter [value= 2.68]is expressed as a hazard ratio, presumably in comparison with healthy subjects).

According to Table 3,  the presence of a screening increases the rate to diagnose a localised disease from 0.181 to 0.702. References 9 and 20 are provided in support of these values. Probably, these parameters A and B are likely to be subjected to a quite limited variability, and anyhow their values reported in Table 3 are quite close to what common sense suggests. Despite this, more explanations would be needed regarding the values of A and B in Table 3 and, more importantly, A and B should be evaluated in the sensitivity analysis. In the current form, Figure 2 is unclear because it does not adequately describe the meaning of  “% of localised LC diagnosed with screening (50%-80%).”

In contrast, there seems to be an important issue with parameters D and E. Apart from the probable mistake discussed at the beginning of this review, employing the values of HR=8.38 for disseminated LC and HR=2.68 for localised disease requires much more extensive explanations (apart from citing reference 20) and, above all, both D and E should be evaluated in the sensitivity analysis.

My conclusion is two-fold: a) all the values of ICER are strongly dependent on which values of A, B, C, and D have been used; b) hence, the sensitivity analyses should be much more focused on the effects determined by variations in these four parameters.

All in all, this paper does not provide sufficient arguments to exclude that the values of ICER could be completely different if  one uses values of A, B, C, D as plausible as those employed in this analysis.

SPECIFIC COMMENTS

Line 26: indicate which calendar years

Line 42: the word QALY is missing

Line 66: this is a critical point; citing reference 8 with no further considerations is not enough.

Lines 127-128: please expand this point at this stage or later on.

Lines 156-167: This is not the typical approach to conduct a sensitivity analysis.

-I do not see where the paper reports the number of simulated patients included in the 10,000 Montecarlo simulations, the length of the time horizon and other basic information on the clinical results of the modelling procedure,

-Table 4 provides the results in terms o ICER; before this, another table would be needed to present the clinical results modelled in the simulated population.

Author Response

Please find attached our response to reviewers. 

Round 2

Reviewer 2 Report

The paper can be accepted.